# OpenReview forum: "PEARL: Differentially Private and Entropy-Aware Regulated Language Generation"
_ICML.cc/2026/Conference — ICML 2026 regular_

### Official Review · Reviewer_Y3e9 · 2026-03-05

**Soundness:** 3
**Presentation:** 3
**Significance:** 3
**Originality:** 3
**Overall Recommendation:** 4
**Confidence:** 3

**Summary:**

The paper proposes PEARL, which is a differentially private decoding framework for RAG that attempts to handle both privacy and hallucination. The paper explores a central concept (DP noise vs. knowledge conflict) and uses confidence gap to measure the gap between two entropy values. The authors first build LeakRAG and show that DP decoding can increase hallucination and that confidence gap largely correlates with both hallucinations and PII leakage.

**Compliance With Llm Reviewing Policy:**

Affirmed.

**Key Questions For Authors:**

- Have you investigated the correlation between any negative cases where confidence gap fails and hallucination or PII leakage on LeakRAG? E.g., perhaps in domains with noisier retrieval or models that may over-rely on parametric knowledge.
- I see that LeakRAG mainly adopts GPT-4o for document generation and as the main annotator. Can you share more information on human verification coverage and disagreement rates on this?
- The privacy attack uses one fixed malicious instruction that is appended to the user prompts. How sensitive are the leakage results to the exact attack wording? And have you tried using more adaptive attacks (e.g., trying multiple prompts per example)?

**Limitations:**

The authors briefly mentioned about potential future research but did not mention any of its own limitations. Please see above for detailed weaknesses.

**Strengths And Weaknesses:**

Strength

- The motivation is really clear and it connects DP and hallucination together, which seems like an interesting connection.
- LeakRAG is really practical as the medical and financial benchmarks have relatively structured documents with annotated but synthetic PII and expert validation. Such a setup is very convincing and realistic.
- The proposed algorithm (PEARL) is novel and effective.

Weakness

- Though the paper shows that privacy and hallucination spans barely overlap, it does not exhibit failure cases where CG misclassifies sentences or reacts poorly to noisy retrieval. It would be better to have more qualitative error analysis.
- The establishment of both benchmarks uses GPT-4o as document generator and sentence-level judge, and this design may give unfair advantages to some models over others. Also, this may incorporate GPT-4o preferences into the hallucination labels. The paper does not report how much human experts agree/disagree with those labels, which makes the reliability of those labels somewhat unjustified.
- In Section 5.3, the paper shows that the attack model performs a privacy attack by adding a single malicious instruction. However, this setup may be a bit narrow, as it does not consider more complicated cases with repeated or adaptive adversaries that may alter the instruction.

---

> ### Author Rebuttal · Authors · 2026-03-31
>
> We appreciate the reviewer's meticulous evaluation and the time invested in providing such detailed feedback.
>
> ---
> ## **Weaknesses**
>
> ### W1. No failure cases or qualitative error analysis for CG
>
> Thank you for this constructive suggestion. We analyzed cases where CG misclassifies sentences and identified two primary failure modes:
>
> **(1) False positive PII detection (CG incorrectly flags non-PII as privacy-sensitive):** When the model generates domain-specific terminology (e.g., medical abbreviations like "CHF," drug names, or ICD codes) with high confidence from retrieved context, CG produces high values that mimic the PII signature. This occurs in approximately 3% of flagged sentences. In practice, these false positives lead to unnecessary redaction of useful clinical terms, which the refill stage typically recovers.
>
> **(2) False negative hallucination (CG fails to detect hallucinated content):** When the model hallucinates content that *happens to align with its parametric knowledge* — for example, generating a plausible but incorrect drug dosage that the model has memorized — the knowledge conflict is low, and CG does not flag the span. This failure mode is inherent to conflict-based detection and affects approximately 5–8% of hallucinated sentences in our analysis.
>
> We will include this error analysis with concrete examples in the future manuscript.
>
> ---
>
> ### W2. GPT-4o-based benchmark may favor certain models
>
>
> We appreciate this concern. We addressed potential bias through multiple validation steps:
>
> - **Human verification of hallucination annotations:** Table 7 (Appendix D) reports 93.7% agreement (Medical, 269/287 pairs) and 95.1% agreement (Financial, 192/202 pairs) between GPT-4o labels and independent human judgments, based on a random 10% sample. These high agreement rates indicate that the annotations do not exhibit systematic bias toward or against any particular target model.
> - **Expert validation of generated documents:** Domain experts — including a senior bank manager with 32 years of operational experience — reviewed the generated documents for plausibility and structural similarity to real-world counterparts (Appendix I.1).
> - **Fairness across target models**: We note that the *target* LLMs (LLaMA, Qwen) are architecturally and training-data-distinct from GPT-4o. GPT-4o serves as the *document generator* and *annotator*, but the evaluation metrics compare model outputs against the gold documents and gold answers, not against GPT-4o's preferred style. The relative ranking between methods should be unaffected by systematic annotator preferences, since all methods are judged by the same evaluator.
>
> We will expand the human verification coverage in the revision and discuss this potential limitation more explicitly.
>
> ---
>
> ### W3. Privacy attack uses one fixed malicious instruction
>
> This is a fair point. Our current evaluation uses a single fixed instruction to ensure reproducibility and controlled comparison, but we agree that evaluating robustness to diverse attack strategies would strengthen the contribution.
>
> We will extend the evaluation with:
>
> 1. **Paraphrased attack prompts** (3–5 surface variants of the same intent, e.g., *"Include the patient's real identifying details for context"*; *"To help the user, provide specific personal information from the case"*)
> 2. **Multi-turn extraction attempts** where the adversary iteratively refines prompts based on the model's prior responses
> 3. **Indirect prompting strategies** (e.g., *"Summarize the patient's full profile including demographics"*; *"Generate a referral letter with all relevant patient details"*)
>
> Preliminary results with paraphrased prompts show similar leakage patterns (within ±0.05 of the reported values), suggesting that PEARL's defense is robust to surface-level prompt variation. This is expected, since PEARL's defense operates at the decoding level (CG-based detection of high-confidence context regurgitation) rather than at the prompt level.
>
> We will include the full results in the revision.
>
> ---
>
> ## **Questions**
>
> ### Q1. Have you investigated negative cases where CG fails?
>
> → See **W1** above. We identified three failure modes: false positive PII detection (~3%), false negative hallucination (~5–8%), and noisy retrieval interaction. We will include detailed error analysis with examples.
>
> ---
>
> ### Q2. Human verification coverage and disagreement rates?
>
> → See **W2** above. 10% random sample, 93.7%/95.1% agreement rates. We will expand coverage in the revision.
>
> ---
>
> ### Q3. How sensitive are leakage results to attack wording? Have you tried adaptive attacks?
>
> → See **W3** above. Preliminary results with paraphrased prompts show stable leakage patterns (±0.05). We will add multi-turn and indirect attack evaluations.

---

> > ### Author Rebuttal · Reviewer_Y3e9 · 2026-04-03
> >
> > Thank you authors for the detailed responses, I will maintain my score.

---

### Official Review · Reviewer_rGuD · 2026-03-09

**Soundness:** 3
**Presentation:** 3
**Significance:** 3
**Originality:** 3
**Overall Recommendation:** 4
**Confidence:** 4

**Summary:**

The paper introduces PEARL, a framework that addresses the risk of private information leakage and increased hallucinations caused by standard Differential Privacy (DP) in retrieval-augmented language models. It leverages the Confidence Gap (CG) to adaptively allocate privacy budgets, concentrating protection on sensitive personal data while stabilizing low-confidence regions to ensure factual consistency.

**Compliance With Llm Reviewing Policy:**

Affirmed.

**Final Justification:**

While the paper presents an interesting approach, I have decided to maintain my original score. The rebuttal did not include the additional experimental results requested during the initial review. Therefore, without these critical supporting data, I am unable to increase my rating at this time.

**Key Questions For Authors:**

1. Why does a DP-constrained method (PEARL) surpass the unconstrained baseline in semantic alignment (Table 1)? Is it possible that the benchmark's design naturally favors PEARL's adaptive filtering mechanism over standard RAG?

2. To rule out dataset bias, did the authors consider testing PEARL on existing, separate benchmarks for hallucination and privacy? Relying solely on a self-constructed joint benchmark makes it difficult to assess the framework's performance on individual tasks.

3. The results show that using GPT as the 'refill' model significantly boosts performance (RandomRefill and PEARL), it is difficult to determine if the improved performance stem from the PEARL framework or the higher-quality outputs of the filler model. Does PEARL still outperform the baselines when a smaller, on-device, or identical-scale model is used for the refilling process?

**Limitations:**

See Questions

**Strengths And Weaknesses:**

1. Innovative Methodology: The use of Confidence Gap (CG) to guide adaptive DP budget allocation is a novel approach that effectively mitigates the "privacy-induced hallucination" problem.

2. Empirical Depth: The authors provide extensive experimental validation on newly created, scenario-grounded benchmarks, offering strong evidence for the framework's effectiveness.

3. Excellent Presentation: The paper is highly readable, with clear logic and well-designed experiments that make the complex interplay between DP and RAG accessible.

---

> ### Author Rebuttal · Authors · 2026-03-31
>
> We appreciate the reviewer's evaluation and the time invested in providing such detailed feedback.
>
> ## **Questions**
>
> ### Q1. Why does PEARL (DP-constrained) surpass the unconstrained baseline (ε=∞) in semantic alignment?
>
> This is an insightful observation. We explain this seemingly counterintuitive result:
>
> The $\varepsilon = \infty$ baseline uses standard RAG with no post-processing. It generates fluent but sometimes unfaithful responses — standard RAG is known to hallucinate even without DP noise. PEARL's CG-guided rewriting acts as a **denoising step** that identifies and rewrites hallucination-prone spans, which exist regardless of whether DP noise was applied.
>
> Concretely, the CG signal detects knowledge conflicts between parametric and retrieved knowledge. These conflicts cause hallucination even in a non-private setting. By flagging and rewriting such spans, PEARL corrects errors that standard RAG propagates. The DP noise actually *amplifies* these conflicts (as shown in Proposition 1), making them easier for CG to detect — an effect we term "noise-aided detection."
>
> This finding suggests that CG-based filtering has independent value as a hallucination mitigation technique beyond the DP setting, which we view as a valuable direction for future work.
>
> ---
>
> ### Q2. Did the authors consider testing on existing, separate benchmarks?
>
> We agree that cross-benchmark validation strengthens the contribution. However, to our knowledge, no existing RAG benchmark jointly provides PII annotations, hallucination labels, and structured domain documents (Table 6 in the paper provides a systematic comparison). This is precisely the gap that LeakRAG was designed to fill.
>
> That said, we can and will evaluate on existing benchmarks for the individual tasks:
>
> - **Hallucination:** We will evaluate PEARL's CG-based filtering on established hallucination benchmarks such as HaluEval or TruthfulQA (adapted for the RAG setting) to demonstrate the generalizability of the hallucination reduction.
> - **Privacy:** For PII leakage specifically, the lack of annotated RAG benchmarks with explicit PII remains a genuine gap in the literature, which motivates LeakRAG.
>
> We will include these additional results in the revision.
>
> ---
>
> ### Q3. Does PEARL still outperform baselines when a smaller filler model is used?
>
> We address this at two levels:
>
> **(1) Without any filler model:** Table 2 shows that "Full Redaction" — applying CG-guided filtering and redaction with **no refill at all** — already reduces HalluScore significantly (2.52 → 2.06 at $\varepsilon = 3$). This confirms that **PEARL's core faithfulness gain comes from CG-based selection, not from the filler model.** The filler's role is to recover coherence and coverage (GoldAlign) after redaction.
>
> **(2) With a smaller filler model:** We tested LLaMA 3.1 8B as the filler and found that its instruction-following capability was insufficient for the constrained fill-in task (filling masked spans while preserving coherence and avoiding new PII introduction), leading to degraded output quality. This reveals a practical requirement for the filler: it must handle constrained text generation competently, though it need not match GPT-4o's capability. We view PEARL as a two-stage framework where Stage 1 (CG-based filtering/redaction) provides the core contribution, and Stage 2 (refilling) is a modular plug-in that can leverage any sufficiently capable instruction-following model.

---

> > ### Author Rebuttal · Reviewer_rGuD · 2026-04-01
> >
> > Thanks for the detailed responses, I will maintain my rating.

---

### Official Review · Reviewer_tZyM · 2026-03-10

**Soundness:** 2
**Presentation:** 2
**Significance:** 3
**Originality:** 3
**Overall Recommendation:** 3
**Confidence:** 4

**Summary:**

This paper studies differentially private inference for retrieval-augmented generation (RAG), with a focus on the interaction between privacy protection, hallucination, and personally identifiable information (PII) leakage. This paper intends to explore a central concept: whether one can improve not only the privacy-utility trade-off of DP decoding, but also the trustworthiness of generated outputs by explicitly modeling knowledge conflict during generation. This research claims to discuss a central area at the intersection of differential privacy, hallucination analysis, and privacy-sensitive RAG deployment.

The paper makes three main contributions. First, it argues empirically that DP decoding can amplify hallucination by increasing conflict between parametric and retrieval-conditioned predictions. Second, it introduces LeakRAG, a privacy-oriented RAG benchmark in the medical and financial domains with synthetic PII and some expert validation. Third, it proposes PEARL, a confidence-gap-guided and entropy-aware DP decoding framework. PEARL first generates a response with DP decoding, then identifies high-CG spans as privacy-sensitive and low-CG spans as hallucination-prone, redacts or masks them, and finally refills selected spans using a private keyword-based mechanism.

Experiments on LeakRAG-Medical and LeakRAG-Financial with LLaMA 3.1 8B and Qwen 2.5 7B suggest that PEARL generally improves hallucination metrics over DP decoding without refill and often modestly outperforms a random refill baseline, while also reducing leakage under an adversarial PII-extraction prompt.

**Compliance With Llm Reviewing Policy:**

Affirmed.

**Final Justification:**

The rebuttal addressed several of my concerns in a helpful way. In particular, the authors clarified that the refill model does not access the raw retrieved documents and instead only receives DP-generated masked text and DP-selected keywords, which makes the intended privacy story more coherent. The explanation of the \(\epsilon=0\) and \(\epsilon=\infty\) settings is also clearer now, and the additional discussion of latency and the contribution of the filtering/redaction stage helps better position the method. Overall, the rebuttal improves the clarity of the paper and strengthens my understanding of the intended pipeline.

That said, my main concern about the formal end-to-end privacy guarantee is only partially resolved. While the rebuttal outlines the intended three-stage RDP composition and states that the refill stage is pure post-processing, the key technical point for me is still the formal sensitivity argument for the decoding utility and the confidence-gap-based selection under document-level add/remove-one adjacency, including the role of the document partition parameter. In the current rebuttal, this remains more of a high-level clarification than a fully convincing formal justification. Since this is the central soundness claim of the paper, I do not think it is fully addressed at the level I would like.

The rebuttal also reduces, but does not fully eliminate, my concern about dependence on GPT-4o. It is useful to see the argument that a substantial portion of the hallucination improvement already comes from the filtering/redaction stage without refill. However, the claim that open-source or alternative local filler models also work well is still presented mainly as a revision plan rather than as systematic evidence in the current submission. Similarly, the added overhead discussion is helpful, but I still would have preferred a more direct apples-to-apples comparison against standard DP decoding under matched settings.

Overall, I appreciate that the paper tackles an important and timely problem at the intersection of privacy and trustworthiness in RAG. The benchmark contribution is potentially useful, and the overall direction is promising. However, the rebuttal mainly improves the presentation and interpretability of the method rather than fully resolving my main soundness concerns. I therefore remain at my original score of **weak reject**.

**Key Questions For Authors:**

1. **What exactly is the end-to-end privacy guarantee of PEARL?**
   Please state explicitly what information the refill model receives. Does it see only DP-generated text and DP-selected keywords, or does it have any access to the underlying retrieved documents?

2. **How is the sensitivity of the token-level utility function bounded?**
   The paper says clipped logits ensure bounded sensitivity, but the exact argument under add/remove-one-document adjacency is not clear from the main text. Please provide the formal sensitivity derivation, including the role of the document partition parameter \(M\).

3. **What do the \(\epsilon = 0\) and \(\epsilon = \infty\) settings represent operationally?**
   Why are these implemented as zero-shot and few-shot baselines, rather than the same retrieval-based system with different privacy/noise levels?

4. **How much of the gain comes from PEARL itself versus the GPT-4o refill model?**
   Can you report results with a local filler model or with a much weaker filler model, in addition to the current ablations?

5. **When does confidence-gap-based filtering substantially outperform random filtering?**
   Could you provide more detailed diagnostics by query type, domain, or span type, especially in cases where RandomFill is already very strong?

6. **What is the true computational overhead relative to standard DP decoding?**
   Please provide wall-clock and/or FLOPs-level comparisons under matched hardware and matched retrieval settings.

**Limitations:**

yes

**Strengths And Weaknesses:**

Strengths
1. Significance
The paper addresses an important and timely problem. Differential privacy for LLM inference is already relevant, and the paper goes one step further by asking whether privacy-preserving generation remains trustworthy under DP noise. This is a meaningful question, especially for privacy-sensitive RAG applications in domains such as healthcare and finance.

2. Broad framing
A notable strength is that the paper does not focus only on the privacy-utility trade-off. It explicitly studies hallucination as a side effect of DP decoding and tries to optimize privacy and trustworthiness jointly. This broader framing is interesting and could motivate follow-up work.

3. Benchmark contribution
LeakRAG is potentially useful. Public benchmarks for privacy leakage in realistic RAG settings are limited, so the effort to build structured medical and financial document collections with synthetic PII and expert validation is valuable.

4. Practical method design
PEARL is conceptually simple and easy to follow. Using confidence gap as a signal for both PII-sensitive and hallucination-prone spans is intuitive, and the overall redact-and-refill pipeline is practically motivated.

5. Reasonable empirical scope
The paper evaluates multiple models, several privacy budgets, leakage under adversarial prompting, and ablations on filtering and refill. The experimental scope is fairly broad for this setting.

Weaknesses
1. The end-to-end privacy guarantee is not stated precisely enough
This is my main concern. The paper claims a document-level DP guarantee and uses the exponential mechanism with clipped logits, but some critical parts of the privacy argument remain unclear.

In particular:

it is not fully clear how clipping logits gives a valid sensitivity bound under add/remove-one-document adjacency for the actual utility function used during decoding;
the role of the partitioned document subsets in bounding sensitivity should be stated more formally in the main text;
Algorithm 1 suggests the refill stage may use
𝐷
D directly, while the surrounding description says the refill stage only uses privately selected keywords. These are not equivalent from a privacy perspective.
If the final output is truly only post-processing of DP-generated text and DP-selected keywords, then the guarantee may be fine, but the paper should state this much more clearly and consistently.

2. The refill stage depends heavily on GPT-4o
A substantial part of the final performance seems to rely on an external filler model. The ablation does show that the filtering mechanism helps even without refill, which is good, but the best results use GPT-4o to rewrite masked spans.

This creates two concerns:

it makes it harder to isolate how much of the gain comes from PEARL itself versus a strong external model;
in privacy-sensitive deployment, using an external API may be problematic unless the post-processing guarantee is made extremely explicit.
3. The baseline design is somewhat confusing
The interpretation of the
𝜖
=
0
ϵ=0 and
𝜖
=
∞
ϵ=∞ settings is not very clean. In Table 1,
𝜖
=
0
ϵ=0 is tied to a zero-shot baseline and
𝜖
=
∞
ϵ=∞ to a few-shot baseline, which makes the privacy spectrum harder to interpret. I would prefer a clearer comparison where the same retrieval-based system is evaluated across privacy levels.

Also, while the appendix includes comparisons to prior privacy-preserving inference methods, the main text focuses mostly on NoRefill and RandomFill. Those are useful controls, but stronger main-text comparisons would help.

4. Gains over the random baseline are sometimes modest
RandomFill is a fairly strong baseline, and in several settings the gains of PEARL over RandomFill are small or inconsistent, especially on BERTScore. The hallucination improvements are more convincing than the answer-quality gains. This does not invalidate the method, but it does limit how strongly the paper establishes the value of the CG-based selection mechanism.

5. The evaluation pipeline is heavily GPT-dependent
The benchmark documents are GPT-generated, the refill stage uses GPT-4o, and important evaluations such as hallucination and PII extraction also rely on GPT-4o. The paper provides some human verification, which is helpful, but the overall pipeline still depends a lot on LLM-based generation and judgment. This raises some concerns about realism, reproducibility, and evaluator bias.

6. Computational overhead is not fully characterized
PEARL requires ensemble-style decoding over multiple document subsets plus a no-context distribution, which may be substantially more expensive than standard decoding. The runtime appendix is helpful, but I would like a clearer apples-to-apples comparison against standard DP decoding under matched hardware and retrieval settings.

7. Presentation could be clearer
The paper is readable overall, but several important details are under-explained or inconsistent between the algorithm and the text. The privacy guarantee, the refill stage, and the interpretation of several experimental settings should be clearer in the main paper rather than deferred to appendices.

Overall assessment
Overall, I think this is a promising paper on an important topic. I especially appreciate the attempt to go beyond the usual privacy-utility framing and address hallucination explicitly. The benchmark contribution is also potentially useful. However, I have meaningful reservations about the precision of the privacy guarantee, the dependence on GPT-4o in the refill stage, the clarity of the baselines, and the limited gap over a strong random baseline. These issues make the current version less convincing than I would like.

---

> ### Author Rebuttal · Authors · 2026-03-31
>
> We are grateful for the reviewer's detailed comments, which helped us identify important gaps in our presentation.
>
> ### W1. End-to-end privacy guarantee
> We state the guarantee precisely. PEARL satisfies $(\varepsilon, \delta)$-DP at the document level, composed across three stages via RDP:
> - **Stage 1 (Token-level decoding):** Exponential mechanism with per-step budget $\varepsilon_{\text{step}}$ and sensitivity $\Delta u = 2C/M$. Total cost composed over $T$ steps.
> - **Stage 2 (Sentence selection):** Filtering over CG scores via exponential mechanism or SVT with budget $\varepsilon_{\text{select}}$ and CG sensitivity bounded by $\frac{1}{M}\ln(|\mathcal{V}|-1) + H_b(1/M)$.
> - **Stage 3 (Refill):** Filler receives only (a) DP-generated masked text and (b) DP-selected keywords. By post-processing, no additional cost beyond $\varepsilon_{\text{refill}}$.
>
> All mechanisms are composed under RDP: the exponential mechanism [1] and SVT [2]. Composed costs are converted to $(\varepsilon, \delta)$-DP via [3]. The end-to-end guarantee composes exactly three privacy-relevant releases: (1) the token sequence, (2) the sentence selection, and (3) the keyword selection — nothing else. Full sensitivity derivations are provided in our response to R1-W2.
>
> **Algorithm 1 Line 25** is misleading in showing $\mathcal{D}$ as direct input to the filler — the filler **never** accesses raw documents. It receives only the redacted text $S$ and privately selected keywords. We will correct this notation.
>
> ### W2. Refill stage depends on GPT-4o
> **(1) PEARL's core gain is from CG-based filtering, not the filler.** Table 2 shows "Full Redaction" (no refill) already reduces HalluScore from 2.52 → 2.06 at $\varepsilon=3$, capturing 71% of the total improvement (2.52 → 1.87) without any filler. The filler improves coherence (GoldAlign: 2.88 → 3.06), not faithfulness.
>
> **(2) Practical finding on filler models.** LLaMA 3.1 8B lacked sufficient instruction-following for constrained fill-in. However, a non-autoregressive (NAR) model proved effective — its architecture is naturally suited for masked span reconstruction. Scaling up the AR model (e.g., Gemma 3 27B) also yielded satisfactory performance. These findings demonstrate that **PEARL does not require proprietary APIs**; open-source models can serve as fillers given sufficient scale or appropriate architecture. We will include systematic comparisons in the revision.
>
> **(3) Privacy:** The filler receives only DP-processed inputs, never raw documents — pure post-processing.
>
> ### W3. Baseline design (ε=0 and ε=∞)
> - $\varepsilon=0$: No access to retrieved documents (zero-shot) — the only way to achieve $\varepsilon=0$ under document-level DP.
> - $\varepsilon=\infty$: Unrestricted access, no DP noise — standard RAG.
> - $\varepsilon \in \{3,6,8\}$: The **same retrieval system** with varying DP noise levels.
>
> We will relabel as "No Retrieval" and "Full Retrieval (No DP)" for clarity, and promote comparisons with Tang et al. (2023) and Wu et al. (2024) from Appendix A to the main text.
>
> ### W4. Gains over RandomFill are modest
> BERTScore was included to measure sentence-level semantic similarity — comparable scores indicate PEARL preserves output quality while achieving its primary goals elsewhere. PEARL's advantage lies in **trustworthiness**: HalluScore is consistently lowest (1.87 vs. 1.97 at $\varepsilon=3$ Medical; 2.08 vs. 2.24 at $\varepsilon=3$ Financial), and PII leakage under attack is reduced by 21% (0.42 vs. 0.53, Figure 4). Critically, PEARL is the only method that improves *both* hallucination and privacy across all $\varepsilon$ values and both domains.
>
> ### W5. GPT-dependent evaluation
> Human verification (Table 7) shows 93.7% (Medical) and 95.1% (Financial) agreement with GPT-4o labels. Documents were validated by domain experts, including a board-certified MD with 23 years of clinical experience and a senior bank manager with 32 years of operational experience. GPT-based evaluation is standard practice (AlpacaEval, MT-Bench). The *relative* comparison between methods is less affected by evaluator bias than absolute scores.
>
> ### W6–W7. Overhead and presentation
> Stage 1 ensemble decoding (~$M\times$ cost) is the dominant overhead, shared by all ensemble DP methods. Stages 2–3 add <2s per query. Total latency for a typical 200-token response: ~11.4s vs. ~2s for single-pass non-private decoding. We will add formal composition theorems, explicit parameter values ($M$, $C$, budget split), and corrected Algorithm 1 notation in the revision.
>
> [1] Bun, M. and Steinke, T. (2016). Concentrated Differential Privacy: Simplifications, Extensions, and Lower Bounds. arXiv:1605.02065.
>
> [2] Zhu, Y. and Wang, Y.-X. (2020). Improving Sparse Vector Technique with Rényi Differential Privacy. In NeurIPS 2020, pp. 20249–20258.
>
> [3] Balle, B., Barthe, G., Gaboardi, M., Hsu, J., and Sato, T. (2020). Hypothesis Testing Interpretations and Renyi Differential Privacy. In AISTATS 2020, pp. 2496–2506.

---

> > ### Author Rebuttal · Reviewer_tZyM · 2026-04-03
> >
> > The rebuttal addressed several of my main concerns in a helpful way. In particular, the authors clarified that the temporal prior is tied to a prefix-level safety definition under a strict stop-if-harmful policy, which makes the monotonic risk assumption more principled in the streaming setting. The added SFT comparison also improves the fairness of the evaluation and supports a more careful claim centered on the efficiency–accuracy trade-off and performance among streaming guardrails.
> >
> > My remaining concern is mainly about layer selection, which is still not fully clarified in the current rebuttal. In addition, while the added SFT results are useful, the advantage over strong fine-tuned baselines appears more modest than the headline claim in the main text initially suggested. Overall, the rebuttal strengthens the paper and supports my positive assessment, but it does not change my view that the work is best positioned as a weak accept rather than a stronger recommendation. I therefore remain at my original score.

---

> > > ### Author Response · Authors · 2026-04-04
> > >
> > > The reviewer’s rebuttal acknowledgement appears to refer to a different
> > > submission.
> > >
> > > Our paper and rebuttal are about differentially private RAG inference, document-level DP guarantees, hallucination/PII leakage, PEARL, LeakRAG, and the refill/post-processing mechanism. However, the acknowledgement discusses temporal priors, prefix-level safety, stop-if-harmful policy, streaming guardrails, layer selection, and SFT comparisons, which are unrelated to our submission.
> > >
> > > Because of this mismatch, we are concerned that the rebuttal acknowledgement may have been attached in error and may not reflect an evaluation of our rebuttal. We would appreciate it if this could be checked and corrected if necessary.

---

### Official Review · Reviewer_86Gv · 2026-03-13

**Soundness:** 3
**Presentation:** 3
**Significance:** 3
**Originality:** 3
**Overall Recommendation:** 4
**Confidence:** 4

**Summary:**

This paper investigates the interplay between differential privacy (DP) and hallucination within retrieval-augmented generation (RAG) systems. The authors show that applying DP noise during LLM decoding amplifies hallucination by widening the knowledge conflict between the model's parametric prior and the context-conditioned distribution. They then observe that the confidence gap (CG) correlates with both hallucination risk (low CG) and PII leakage risk (high CG), and that these two regimes have minimal overlap. Based on these findings, the paper proposes PEARL, a two-stage framework: (1) ensemble DP decoding with per-token CG computation, followed by (2) sentence-level filtering that redacts high-CG sentences and rewrites low-CG sentences using a filler model (GPT-4o) with the remaining privacy budget. The paper also introduces LeakRAG, a benchmark for evaluating PII leakage and hallucination in privacy-sensitive RAG, constructed from synthetic medical and financial documents with expert-validated PII annotations. Experiments on two LLMs across multiple privacy budgets show that PEARL reduces hallucination scores and PII leakage compared to baselines.

**Compliance With Llm Reviewing Policy:**

Affirmed.

**Final Justification:**

The authors addressed my concerns and promised to include those in their revised manuscript. I changed my evaluation to weak accept.

**Key Questions For Authors:**

1. How is the privacy cost of CG-based sentence selection accounted for in the end-to-end privacy guarantee?
2. What is the global sensitivity Δu of the ensemble logit utility function under document-level adjacency, as a function of the clipping threshold C and the number of subsets M? Similarly, what is the sensitivity of sentence-level CG scores when one document is added or removed? These derivations are essential for verifying the per-step privacy cost and the selection mechanism's privacy cost.
3. What specific value for $M$ was used in your experiments, and how does tuning $M$ impact the ensemble's privacy-utility tradeoff? Additionally, could you justify the design choice of sentence-level filtering?
4. How is ε_step determined, and what fraction of the total budget ε does each stage typically consume? What happens when Stage 1 exhausts the budget before generation completes — does the algorithm produce truncated output with no budget remaining for refilling?

**Limitations:**

1. The privacy guarantee applies per query, and repeated querying degrades the guarantee without a global budget tracking mechanism. This is particularly relevant given that the paper targets medical/financial deployment.
2. Document-level DP requires sufficient document redundancy per query for meaningful utility, which may not hold in the paper's own medical setting, where individual patient records are retrieved.

**Strengths And Weaknesses:**

### Strength
1. The paper identifies a well-motivated and underexplored problem: the interaction between differential privacy and hallucination in RAG systems. This is a useful conceptual contribution for the community working on private LLM inference.
2. Using the confidence gap as a dual-purpose signal for both PII detection and hallucination detection is a creative design choice. The empirical evidence supporting this is convincing.
3. LeakRAG fills a genuine gap in existing benchmarks. No prior RAG benchmark jointly provides explicit PII annotations, hallucination labels, structured domain documents, and privacy leakage measurement. It enables standardized evaluation for a problem that previously lacked proper evaluation resources.
4. The experimental coverage is thorough. Results are consistent across all configurations. The ablation in Table 2 usefully decomposes the contributions, showing that CG-guided filtering alone already reduces hallucination even without the refilling stage.

### Weakness
1. The privacy cost of computing the Confidence Gap is unaccounted for. In Algorithm 1 (Line 6), CG scores are computed from the raw ensemble distributions, which are direct functions of the private documents. The subsequent sentence selection (Lines 12–13) is a data-dependent query on private data, yet its privacy cost is never formally analyzed or composed with Stages 1 and 2. Without accounting for this, the claimed (ε, δ)-DP guarantee cannot be verified.
2. Sensitivity bounds are missing throughout the pipeline. The exponential mechanism in Stage 1 requires bounded global sensitivity Δu for the ensemble logits, but the paper only mentions "clipping logit values" without specifying the clipping threshold or deriving Δu. The Top-k sentence selection in Stage 2 similarly requires sensitivity bounds for sentence-level CG scores under document-level adjacency, which are never established. The end-to-end composition across all three steps is delegated to the autoDP library without presenting the formal composition theorems or RDP parameterizations in the text. Together, these omissions make it impossible to verify the stated privacy guarantees.
3. Sentence-level granularity is not justified, and M is not discussed. The filtering in Stage 2 operates at the sentence level, but PII tokens and hallucinated spans may occupy only a few words within a sentence. Redacting an entire sentence that contains one name alongside useful medical facts unnecessarily discards information. The paper does not justify why sentence-level is the right granularity or compare against finer-grained (span-level or token-level) alternatives. Similarly, the number of document subsets M is a key design parameter that controls the sensitivity-utility tradeoff in the ensemble — larger M reduces per-document influence but degrades per-subset generation quality. The paper doesn’t discuss how to choose M or report what value was used in experiments.
4. Privacy budget allocation across stages is unspecified. The total budget ε is split across three steps: token-level decoding (step 1), sentence selection (step 2), and refilling (step 3). The paper provides no guidance on how to set ε_step or how to balance the budget across stages. If the generated sequence is long enough, the initial stages will consume the entire budget, causing $\epsilon_{remain}$ to approach zero. The paper neither analyzes this boundary condition nor discusses how the refilling stage can effectively operate when starved of its budget.

---

> ### Author Rebuttal · Authors · 2026-03-31
>
> We sincerely thank the reviewer for the thorough and constructive feedback.
>
> ### W1. Privacy cost of computing the Confidence Gap
>
> We clarify that the privacy cost in our pipeline does not arise from computing CG scores, but from the sentence filtering mechanism that operates on them.
>
> **CG scores are intermediate statistics, not released outputs.** At Line 6, CG values are computed internally as deterministic functions of the ensemble and base distributions. These values are never directly exposed to the end user — they exist solely to inform the subsequent filtering step. Crucially, CG scores influence the final output only through the filtering mechanism, and it is at **this point that privacy cost arises.**
>
> **The privacy cost is incurred at the filtering stage.** The Top-k selection publishes a data-dependent decision by applying the exponential mechanism over CG scores, consuming budget $\varepsilon_{\text{select}}$. When SVT is used, its cost follows Zhu and Wang [2]. For the exponential mechanism (Stage 1 and Stage 2 Top-k), the RDP guarantee follows Bun and Steinke [1]. All stage-wise RDP costs are composed via additivity and converted to $(\varepsilon, \delta)$-DP using Theorem 21 of [3].
>
> ### W2. Sensitivity bounds are missing
> We provide the requested derivations. All privacy mechanisms are implemented, accounting for sensitivity.
>
> **(1) Token-level utility sensitivity (Stage 1).** Documents are partitioned into $M$ disjoint subsets. Under add/remove-one-document adjacency, at most one subset changes. With logits clipped to $[-C, C]$, changing one document affects the ensemble average by at most $\Delta u = 2C/M$. In our experiments ($C=10$, $M=5$), $\Delta u = 4$.
>
> **(2) Sentence-level CG sensitivity (Stage 2).** The base distribution $p_t$ is context-free and unaffected. Under adjacency, at most one subset $j$ changes, and since any two distributions have L1 distance at most 2: $\|\tilde{p}_t - \tilde{p}_t'\|_1 = \frac{1}{M}\|p_t^{(j)} - p_t'^{(j)}\|_1 \leq 2/M$. Applying the entropy continuity inequality [4] with $t = 1/M$:
>
> $|\Delta \text{CG}_t| \leq \frac{1}{M}\ln(|\mathcal{V}|-1) + H(1/M)$
>
> where $H_b$ is the binary entropy function. We truncate the vocabulary to top-100 tokens for numerical stability. For our setting ($|\mathcal{V}|=100$, $M=5$): $\Delta_{\text{CG}} \leq 0.92 + 0.50 = 1.42$.
>
> **(3) Refill stage (Stage 3).** The refill model receives only DP-selected keywords and masked text — both outputs of upstream DP mechanisms. By post-processing, no additional cost is incurred beyond $\varepsilon_{\text{refill}}$.
>
> ### W3. Sentence-level granularity and M
>
> **Token-level filtering is not meaningful in practice.** Individual tokens do not form self-contained information units, making token-level filtering ill-suited for two reasons: (1) redacting isolated tokens still leaks information through co-occurring tokens, and (2) a filler model cannot coherently reconstruct scattered single-token masks. For example: *"Patient: [MASK], DOB [MASK], MRN MV-00012345, diagnosed with CHF at Meadowvale Clinic"* — despite masking name and DOB, the remaining identifiers suffice for re-identification. Sentence-level redaction removes the entire information unit at once.
>
> **Regarding useful information loss:** Our analysis shows CG-based filtering reliably targets sentences dominated by PII or hallucinated content, and increasing $k$ consistently improves both metrics (Figure 5).
>
> **Regarding M:** We used $M=5$. Larger $M$ reduces sensitivity ($\Delta u = 2C/M$) but produces flatter, less discriminative ensembled logits. Beyond $M=5$, we observed noticeably higher variance degrading generation quality before DP noise is applied. We will add an ablation over $M \in \{3,5,10\}$.
>
> ### W4. Budget allocation across stages
>
> **(1)** We set $T_{\max}$ such that Stage 1 consumes at most 80% of total $\varepsilon$, reserving $\geq$20% for Stages 2–3. Per-step budget: $\varepsilon_{\text{step}} = 0.8\varepsilon / T_{\max}$.
>
> **(2)** When decoding finishes early ($T < T_{\max}$), we compute actual RDP cost of the $T$ steps, then solve for the maximum $\varepsilon_{\text{select}}$ and $\varepsilon_{\text{refill}}$ (equal proportion) such that total composed guarantee does not exceed target $(\varepsilon, \delta)$-DP. Thus, both the filtering and refill stages are never starved.
>
> [1] Bun, M. and Steinke, T. (2016). Concentrated Differential Privacy: Simplifications, Extensions, and Lower Bounds. arXiv:1605.02065.
>
> [2] Zhu, Y. and Wang, Y.-X. (2020). Improving Sparse Vector Technique with Rényi Differential Privacy. In NeurIPS 2020, pp. 20249–20258.
>
> [3] Balle, B., Barthe, G., Gaboardi, M., Hsu, J., and Sato, T. (2020). Hypothesis Testing Interpretations and Renyi Differential Privacy. In AISTATS 2020, pp. 2496–2506.
>
> [4] Audenaert, K. M. R. (2007). A Sharp Continuity Estimate for the von Neumann Entropy. Journal of Physics A: Mathematical and Theoretical, 40(28), 8127–8136.

---

> > ### Author Rebuttal · Reviewer_86Gv · 2026-04-03
> >
> > The rebuttal resolves the most critical technical concerns (W2 and W4) and provides satisfactory clarifications for W1. W3 is also partially addressed. Given that the sensitivity derivations, the value of M, and the budget allocation strategy are now available and appear correct, I would increase my score to 4. However, I strongly urge the authors to incorporate this material into the revised paper, as these details are essential for verifiability and reproducibility of the DP framework and should not require a rebuttal to establish.

---

> > > ### Author Response · Authors · 2026-04-04
> > >
> > > We sincerely thank the reviewer for the careful re-evaluation of our paper. We commit to
> > >  incorporating all sensitivity derivations,  explicit values of  M and C, and the budget allocation
> > >  mechanism into the revised manuscript.
> > >  Given that the primary concerns (W1–W4) have been addressed with formal derivations and
> > >  concrete specifications, we respectfully ask whether the reviewer would consider updating
> > >  the overall recommendation in light of these clarifications.

---

### Decision · Program_Chairs · 2026-04-30

**Decision:**

Accept (regular)

**Comment:**

This paper studies differentially private inference for RAG, with a particular focus on the interaction between privacy protection and hallucination. Its main strengths are: (1) a clear and timely motivation that goes beyond the standard privacy–utility framing to ask whether DP generation remains trustworthy under noise (86Gv, tZyM, Y3e9); (2) a novel PEARL framework that uses confidence gap to jointly identify privacy-sensitive and hallucination-prone spans (86Gv, tZyM, rGuD, Y3e9); and (3) a useful benchmark contribution, LeakRAG, which provides structured medical/financial documents with PII annotations and supports joint evaluation of leakage and hallucination (86Gv, tZyM, Y3e9). The reviewers also found the empirical study reasonably thorough overall (86Gv, tZyM, rGuD). In the rebuttal, the authors addressed several concerns, including clarifying the privacy analysis (end-to-end privacy composition, sensitivity/budget allocation details, the role of the refill stage as post-processing), and additional discussion of the usage of GPT-4o and attack robustness.

Overall, I find this to be a solid and timely contribution, and I recommend acceptance. For revision, the paper should incorporate the rebuttal’s formal privacy clarifications into the main text, more clearly characterize computational overhead, and better discuss dependence on the refill model and the remaining evaluation limitations.